# Rapid Effect of Benralizumab for Hypereosinophilia in a Case of Severe Asthma with Eosinophilic Chronic Rhinosinusitis

**DOI:** 10.3390/medicina55070336

**Published:** 2019-07-03

**Authors:** Hiroaki Tsurumaki, Toshiyuki Matsuyama, Kazuma Ezawa, Yasuhiko Koga, Masakiyo Yatomi, Haruka Aoki-Saito, Kazuaki Chikamatsu, Takeshi Hisada

**Affiliations:** 1Department of Respiratory Medicine, Gunma University Graduate School of Medicine, Maebashi 371-8511, Japan; 2Department of Otolaryngology-Head and Neck Surgery, Gunma University Graduate School of Medicine, Maebashi 371-8511, Japan; 3Gunma University Graduate School of Health Sciences, Maebashi 371-8514, Japan

**Keywords:** severe asthma, benralizumab, hypereosinophilia, bronchial thermoplasty, eosinophilic chronic rhinosinusitis

## Abstract

A 56-year-old man with severe asthma underwent bronchial thermoplasty (BT). However, his asthma exacerbated and hypereosinophilia developed 2 months later, thus necessitating oral corticosteroid (OCS) therapy. Six months after BT, a diagnosis of severe asthma with eosinophilic chronic rhinosinusitis (ECRS) was made and benralizumab treatment was initiated; the blood eosinophil count subsequently decreased and lung function improved, thereby permitting OCS dose tapering. Surprisingly, benralizumab both reduced nasal polyps and ameliorated ECRS. Thus, benralizumab may be a useful drug for the rapid treatment of severe asthma with ECRS, especially in patients with hypereosinophilia.

## 1. Introduction

Severe asthma is defined as asthma that remains uncontrolled despite step 4 or 5 treatment, according to the Global Initiative for Asthma (GINA) guidelines [1]. Recommended GINA step 5 treatment involves the use of high-dose inhaled corticosteroids, long-acting β2-agonist, leukotriene-receptor antagonist, and add-on treatments, including biologics. In addition, bronchial thermoplasty (BT) is considered a potential treatment option for select patients with severe asthma.

Benralizumab is a monoclonal antibody against interleukin (IL)-5 receptor subunit α (IL-5Rα), which depletes eosinophils via antibody-dependent cell-mediated cytotoxicity (ADCC) [2]. In the SIROCCO and CALIMA trials, which were randomized, multicenter, placebo-controlled, phase 3 trials involving patients who had severe, uncontrolled asthma with eosinophilia, the investigators found that, compared to placebo, benralizumab reduced the annual asthma exacerbation rate and significantly improved prebronchodilator forced expiratory volume in 1 s (FEV1) [3,4].

Many patients with severe asthma have comorbidities (e.g., allergic rhinitis and eosinophilic chronic rhinosinusitis (ECRS)); notably, ECRS is characterized by the presence of bilateral refractory chronic rhinosinusitis with nasal polyps, dominant ethmoid sinus shadows on computed tomography (CT) images, and eosinophilic infiltration [5]. ECRS is an important factor influencing asthma control. Fractionated expiratory nitric oxide (FeNO) levels correlate with eosinophilic airway inflammation in asthma and are elevated in uncontrolled asthma [6] and/or comorbidities, including allergic rhinitis, rhinosinusitis, and gastroesophageal reflux disease [7,8]. Notably, patients with well-controlled asthma exhibit elevated FeNO levels when ECRS is also present, but not when ECRS is absent [9]. This indicates that both asthma and ECRS are closely associated and occur as a single airway disease. Therefore, benralizumab may be useful for managing severe asthma with ECRS. Here, we describe a 56-year-old man with severe asthma and ECRS who developed hypereosinophilia and exhibited a rapid response to benralizumab treatment.

## 2. Case Presentation

A 51-year-old man (written inform consent received) diagnosed with bronchial asthma experienced frequent asthma exacerbations over the course of 5 years after diagnosis, with oral corticosteroid (OCS) burst therapy required for the management of these exacerbations. The patient was treated with omalizumab, a monoclonal antibody against immunoglobulin (Ig) E, because his total IgE level and the specific IgE levels of house dust mite and Japanese cedar were high (Table 1); however, his asthma was exacerbated frequently. His FeNO level remained above 100 ppb, while his forced expiratory volume % in 1 s (FEV1%) decreased with time, despite GINA step 5 therapy.

The patient underwent BT 5 years after the diagnosis of asthma and experienced improvements in the scores of asthma quality-of-life questionnaire (AQLQ) and FEV1% (Figure 1) [10]. Two months after BT, however, his asthma exacerbated and blood eosinophil count and FeNO level increased, necessitating OCS therapy. The hypereosinophilia became aggravated when the OCS dose was tapered; consequently, a high dose of 10 mg/day or greater was required for the control of asthma symptoms and hypereosinophilia (Figure 1). Six months after BT, FEV1% exhibited a slight decrease, and the patient experienced both nasal congestion and hyposmia. CT revealed dominant ethmoid sinus shadows (Figure 2A), and the Lund–Mackey score for chronic rhinosinusitis was 6 [11]. Endoscopy revealed nasal polyps, and pathological examination showed eosinophilic infiltration (eosinophil count in the nasal polyps was 215–369 per high-power field (Figure 2B,C). A diagnosis of severe asthma with ECRS was made and benralizumab treatment was initiated. The patient received 30 mg of benralizumab by subcutaneous injection once every 4 weeks for the first three doses, followed by an injection every 8 weeks thereafter. During the initial 4 weeks of treatment, the blood eosinophils were completely depleted, and both AQLQ and FEV1% scores increased (Figure 1). Surprisingly, at 16 weeks after treatment initiation, the ethmoid sinus shadows observed on CT images (Figure 2D) had resolved. The nasal polyps (Figure 2E) had reduced in size; moreover, eosinophilic infiltration in nasal polyps was reduced at 1 year after treatment initiation (eosinophil count in nasal polyps was 0–2 per high-power field; Figure 2F), with improved nasal visual analog scale (VAS) scores. The CT scores improved from 6 to 2, while the nasal VAS scores improved from 5 to 0. Moreover, the OCS dose could be gradually tapered from 30 mg/day to 3 mg/day by 16 weeks.

## 3. Discussion

In this report, we described the effectiveness of benralizumab treatment for hypereosinophilia in a patient with severe asthma and ECRS.

Our patient experienced frequent exacerbations despite GINA step 5 therapy, including omalizumab. Currently, we can select mepolizumab, a monoclonal antibody against IL-5, or benralizumab for the treatment of patients who have high peripheral eosinophil counts [12,13]. However, we could not select these antibodies for treatment of the patient in this case, because both mepolizumab and benralizumab were unavailable at that time. The patient underwent BT and experienced improvements in AQLQ scores and FEV1% after BT. In the AIR trial, the morning peak expiratory flow, symptoms, and quality of life exhibited significantly greater improvements in the BT group than in the control group [14]. Subsequently, the 5-year effectiveness of BT with regard to asthma control and safety was reported in the AIR2 trial [15,16].

Benralizumab was effective for both bronchial asthma and ECRS in the present case. In particular, peripheral blood eosinophils were completely depleted only 4 weeks after the initiation of treatment. In addition, the ethmoid sinus shadows on CT images and nasal VAS scores showed rapid improvement. Previous studies have shown that mepolizumab restored the peripheral blood eosinophil count to a normal level and did not allow further exacerbation of asthma [12,17]. In other studies, mepolizumab reduced the nasal polyp size, although the required dose was more than seven times the normal dose for bronchial asthma [18,19]. Surprisingly, the same dose of benralizumab ameliorated both bronchial asthma and ECRS in our patient. Previous studies showed that benralizumab lowered the peripheral blood eosinophil count to an undetectable level, and that it was more effective in the high eosinophil count group than in the low eosinophil count group [3,4,13,20]. Moreover, a decrease in the eosinophil count in the airway mucosa/submucosa and sputum was found after benralizumab administration [2]. Similarly, in the present case, benralizumab resolved ECRS by depleting the highly concentrated eosinophils in the peripheral blood and in the tissues of the ethmoid sinuses and nasal polyps. These findings indicate that benralizumab, which binds with high affinity to IL-5Rα, depletes eosinophils in both peripheral blood and in tissues by inducing apoptosis through enhanced ADCC.

The patient exhibited exacerbation of asthma with hypereosinophilia at 6 months after BT required treatment with an increased dose of OCS. During OCS dose tapering, a diagnosis of severe asthma with ECRS was made. Anti-IL-5 or IL-5Rα therapy was not available before the patient underwent BT. With currently available treatments, it would be ideal for this patient to receive anti-IL-5 or IL-5Rα therapy before BT, rather than after BT. Overall for treatment of patients who have asthma with hypereosinophilia, priority should be given to anti-IL-5 or IL-5Rα therapy, rather than BT. This report demonstrated that benralizumab improved asthma control and resolved the nasal polyps and ethmoid sinus shadows on CT images of ECRS by depleting eosinophils in the peripheral blood, as well as in nasal polyp tissues. Benralizumab appears effective for nasal polyps and ECRS, and several clinical trials are underway for severe chronic rhinosinusitis with eosinophilic polyposis.

## 4. Conclusions

Benralizumab may be effective for controlling high eosinophil counts in both blood and tissues; thus, it can result in rapid improvement of hypereosinophilia in patients who have severe asthma with ECRS.

## Figures and Tables

**Figure 1 medicina-55-00336-f001:**
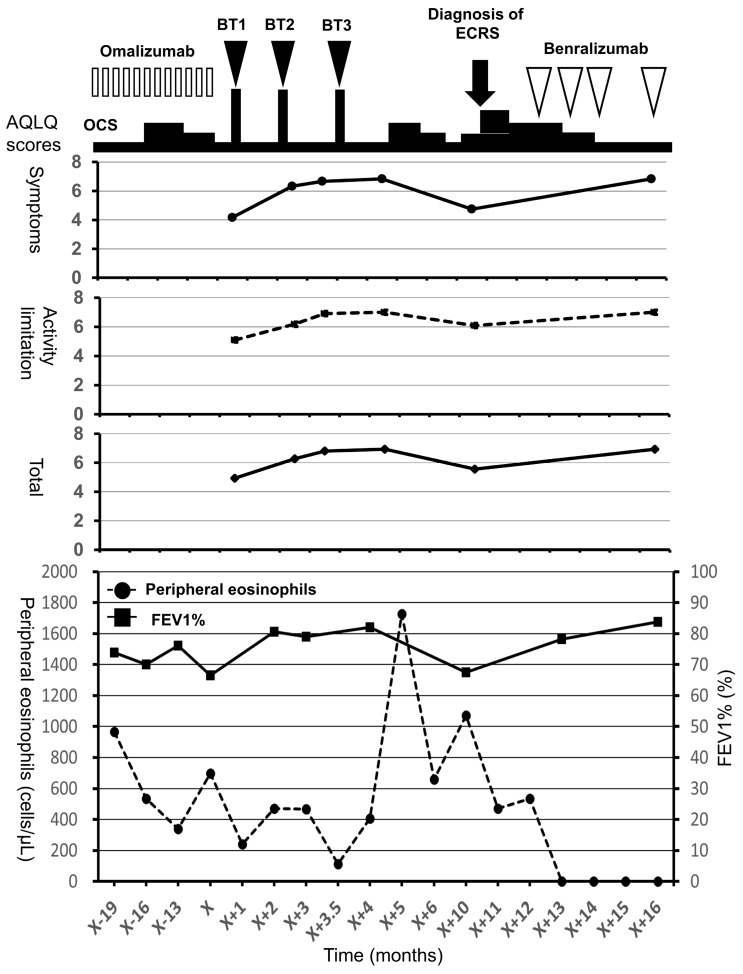
Clinical course and asthma quality-of-life questionnaire scores for a 56-year-old man with severe asthma and eosinophilic chronic rhinosinusitis. In the top panel, the open bars indicate treatment with omalizumab, the closed arrow heads indicate treatment with BT, and the open arrow heads indicate treatment with benralizumab. In the middle panel, the closed circles with the straight line indicate the symptom scores, the closed squares with the broken line indicate the activity limitation scores, and the closed rhombuses with the straight line indicate total scores. In the bottom panel, the closed circles with the broken line indicate peripheral eosinophils and the closed squares with the straight line indicate FEV1%. AQLQ, asthma quality-of-life questionnaire; BT, bronchial thermoplasty; ECRS, eosinophilic chronic rhinosinusitis; FEV1%, forced expiratory volume % in 1 s; OCS, oral corticosteroids.

**Figure 2 medicina-55-00336-f002:**
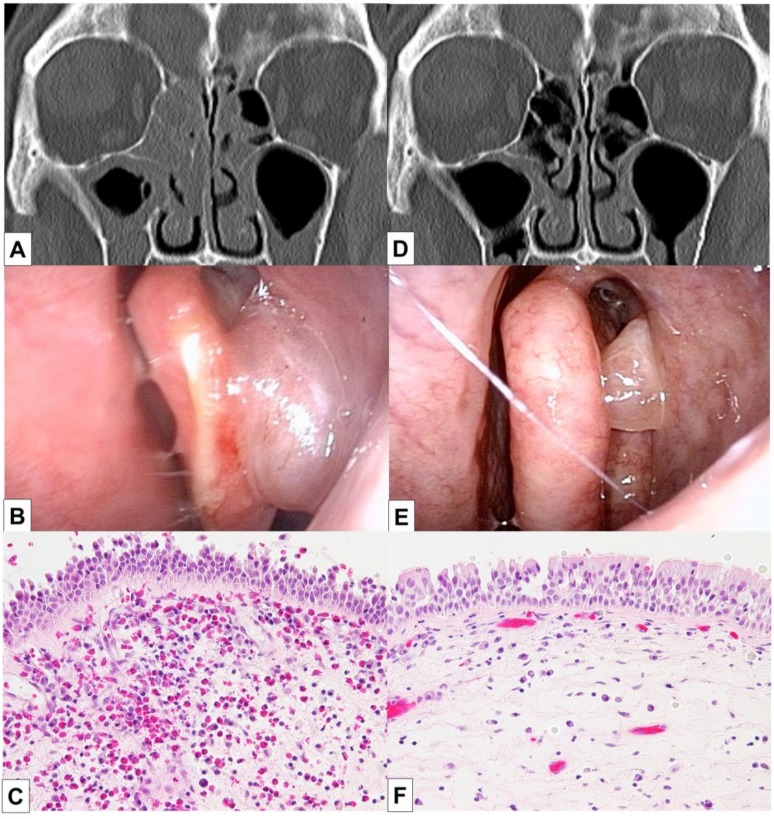
Findings of endoscopy, CT, and pathological analyses in a 56-year-old man with severe asthma and eosinophilic chronic rhinosinusitis who developed hypereosinophilia after bronchial thermoplasty. The CT images show ethmoid sinus shadows, while nasal polyps are visible on endoscopy and histopathology analyses of biopsy samples from the nasal polyps. The images in (**A**–**C**) were obtained before benralizumab treatment, while those in (**D**,**E**) were obtained after 16 weeks of benralizumab treatment. The image in (**F**) was obtained after 1 year of benralizumab treatment; it shows reduced accumulation of eosinophils in the nasal polyps. Benralizumab treatment resolved the ethmoid sinus shadows, reduced nasal polyp size, and decreased eosinophilic infiltration into the nasal polyps. CT, computed tomography.

**Table 1 medicina-55-00336-t001:** Laboratory data.

Laboratory Data Before the Treatment of BT								
Hematology			Chemistry					
WBC	10,400	/μL	TP	7.1	g/dL	BUN	13	mg/dL
Neu	7150	/μL	Alb	4.3	g/dL	Cr	1.02	mg/dL
Eos	690	/μL	T-bil	1.2	mg/dL	Na	141	mEq/L
Bas	110	/μL	AST	19	U/L	K	4.1	mEq/L
Mon	290	/μL	ALT	24	U/L	Cl	106	mEq/L
Lym	2110	/μL	LDH	268	U/L	Ca	9.6	mEq/L
RBC	552	×10^4^/μL	ALP	267	U/L	BNP	5.1	pg/mL
Hb	17.5	g/dL	GGTP	32	U/L	CRP	0.19	mg/dL
Plt	18.7	×10^4^/μL	CK	344	U/L	Glu	150	mg/dL
			UA	6.8	mg/dL	HbA1c	6.1	%
Coagulation			T-chol	161	mg/dL			
PT	104	%	HDL-C	47	mg/dL	Serology		
APTT	28.5	sec	LDL-C	91	mg/dL	total IgE	457.3	IU/mL
D-dimer	0.5	μg/mL	TG	180	mg/dL			
Serology data before the treatment of omalizumab								
total IgE	267	IU/mL	house dust mite		8.83	UA/mL (Class 3)		
			Japanese cedar		4.68	UA/mL (Class 3)		

BT, bronchial thermoplasty; WBC, white blood cells; RBC, red blood cells; Hb, hemoglobin; Plt, platelets; PT, prothrombin time; APTT, activated partial thromboplastin time; TP, total protein; Alb, serum albumin; T-bil, total bilirubin; AST, aspartate aminotransferase; ALT, alanine aminotransferase; LDH, lactate dehydrogenase; ALP, alkaline phosphatase; GGTP, gamma-glutamyl transpeptidase; CK, creatine kinase; UA, uric acid; T-chol, total cholesterol; HDL-C, high density lipoprotein cholesterol; LDL, low density lipoprotein cholesterol; TG, triglyceride; BUN, blood urea nitrogen; Cr, creatinine; BNP, brain natriuretic peptide; CRP, C-reactive protein; Glu, glucose, HbA1c, hemoglobin A1c; IgE, immunoglobulin E.

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
