# Peer review of "Rapid Effect of Benralizumab for Hypereosinophilia in a Case of Severe Asthma with Eosinophilic Chronic Rhinosinusitis"

_medicina, 2019, doi:10.3390/medicina55070336_

Reviewer 1 Report

Tsurumaki et.al. report that Patient undergoes BT after 5 years of diagnosis of asthma. During this period he has received omalizumab. BT an invasive procedure exacerbated asthma and lead to hypereosinophilia (ECRS). OCS therapy also did not work. Benralizumab therapy which targets the IL5 receptor alpha subunit, lead to complete resolution of the peripheral eosinophilia. Though study does present that benralizumab is effective agianst aggressive eosinophilia, I am afraid the claims made in the paper are not novel enough and present addition to growing body of evidences supporting the use of benralizumab effectiveness in treating hypereosinophila.

 First of all, it has already presented in the benralizumab in the case of eosinophilia see https://doi.org/10.1016/j.anai.2018.09.422, 10.2147/TCRM.S157171 and several others. Study comes at no surprise, thus final conclusion is just a version of already published results.

 Figure2 also show that peripheral eosinophil counts are though increased right after BT therapy, and also tapering followed by that, Benralizumab resolved it quickly than its anticipated course, which was slower with OCS. However, at the time of assigning diagnosis of the ECRS has eosinophil count just as much as in the beginning of the very first measurement before BT, was it missed all along in the very first place?

Author Response

Response to review 1 comments

We wish to express our gratitude to the reviewer for the insightful comments. They have helped us significantly improve the paper. We have responded to the comments below

Point 1: First of all, it has already presented in the benralizumab in the case of eosinophilia see https://doi.org/10.1016/j.anai.2018.09.422, 10.2147/TCRM.S157171 and several others. Study comes at no surprise, thus final conclusion is just a version of already published results.

Response 1: We appreciate to the reviewer for the comment. The article on eosinophilic esophagitis (https://doi.org/10.1016/j.anai.2018.09.422, 10.2147/TCRM.S157171) is interesting and important in case selection of treatment with benralizumab. We agree that benralizumab is effective for hypereosinophilia with bronchial asthma, eosinophilic esophagitis, and other eosinophilic diseases. However, we believe that the effectiveness of benralizumab for bronchial asthma with ECRS would be high because peripheral eosinophils completely depleted with benralizumab treatment in our case. Furthermore, eosinophils in the nasal polyps also depleted. We have added pathological images of the nasal polyps before the initiation of benralizumab in Figure 2C and 1 year after initiation of benralizumab in Figure 2F in the revised manuscript). It is a novel finding that benralizumab resolved ECRS by depleting the highly concentrated eosinophils in the peripheral blood and tissues of the ethmoid sinuses and nasal polyps.

Point 2: Figure2 also show that peripheral eosinophil counts are though increased right after BT therapy, and also tapering followed by that, Benralizumab resolved it quickly than its anticipated course, which was slower with OCS. However, at the time of assigning diagnosis of the ECRS has eosinophil count just as much as in the beginning of the very first measurement before BT, was it missed all along in the very first place?

Response 2: In accordance with the reviewer’s comment, this patient might have ECRS before initiation of BT; however, at the time of initiation of BT, he did not have any nasal symptoms associated with ECRS. Furthermore, the count of peripheral eosinophils at the time of initiation of BT were equal to that at the time of ECRS diagnosis. However, the OCS dose administrated before BT (2.5 mg per day of prednisolone) was different from that administered at the time of ECRS diagnosis (10 mg per day). Therefore, CT of the ethmoid sinuses should be performed to diagnose ECRS.

   The anti-IL-5 or IL-5R therapy was not available at the time of treatment with BT. He was eosinophilic, and general oral corticosteroids were needed to control asthma symptoms. He also had frequent exacerbations, which were treated with “oral corticosteroid burst.” We selected BT for this patient, as it had reduced the exacerbation rates in the AIR and AIRII trials. Currently, it would be better for this patient to start an anti-IL-5 or IL-5R therapy before BT rather than after.

   We have added the following sentences in the discussion section (L130–140 in revised manuscript): “The patient had exacerbation of asthma with hypereosinophilia six months after BT initiation, so the oral corticosteroid dose needed to be increased. During the tapering of corticosteroids, severe asthma with ECRS was diagnosed. An anti-IL-5 or IL-5Rα therapy was not available before treatment with BT. Currently, it would be better for the patient to undergo the anti-IL-5 or IL-5R therapy before BT rather than after. To treat patients with eosinophilic asthma, we should prioritize the anti-IL-5 or IL-5Rα therapy over BT. Moreover, benralizumab improved asthma and resolved the nasal polyps and the ethmoid sinus shadows on CT of ECRS by depleting eosinophils from the peripheral blood and nasal polyp tissues. Benralizumab would be effective for nasal polyps and ECRS, and several clinical trials for severe chronic rhinosinusitis with eosinophilic polyposis are in progress.”

  Thank you again for your comments on our paper. I believe that the revised manuscript is suitable for publication.  

Reviewer 2 Report

In this case report by Tsurumaki et al., the authors have reported the effectiveness of benralizumab treatment on 56-year-old man with severe asthma who underwent bronchial thermoplasty. They also went on to conclude their observation that benralizumab reduced nasal polyps and ameliorated eosinophilic chronic rhinosinusitis. The authors ultimately recommend the benralizumab treatment for asthma patients especially with hyper-eosinophilia. The authors duly acknowledged the previous results in support of the treatment. I believe that the study will help improve the care for asthma patients and highly recommend its publication after a minor modification addressing the concerns below.

1.       The X axis of figure 1 lacks the labeling.

2.       Reference article 5 has doi information but others are not. Should there be a uniformity in the reference section?

Author Response

Response to reviewer 2 comments

We wish to express our gratitude to the reviewer for the insightful comments. They have helped us significantly improve the paper.

We have responded to the comments below.

 Point 1: The X axis of figure 1 lacks the labeling.

Response 1:  In accordance with the reviewer’s comment, we added the required label below the X-axis of figure 1. After reflecting on the other reviewer’s comment, we have combined figures 1 and 2 to make a new figure 1 and changed its legends.

 Point 2:  Reference article 5 has doi information but others are not. Should there be a uniformity in the reference section?  

Response 2: In accordance with the reviewer’s comment, we deleted the doi information of the reference article 5.

  Thank you again for your comments on our paper. I believe that the revised manuscript is suitable for publication.

Reviewer 3 Report

This is an interesting case but it would be interesting to expand on the following:

(1) The patient was always eosinophilic so why wasn't aIL5 therapy used instead of BT - perhaps it would have been better to have waited until it was available

(2) Most - up to 90% of asthmatics have co-existent CRS, there was no mention of baseline assessment of CRS in this patient.

(3)Was there a reason for the acute development of hypereosinophila after BT. Could this be a complication of BT?

(4) the conclusion states that tissue eos were reduced but there was no post Rx biopsy data reported

(5)Figs 1 and 2 there is a lot of duplication in descriptive wording.

(6) There is existing data to show that CRSwNP responds well to benralizumab suggesting that this is not a novel finding (Ref: Bleecker Er et al Baseline patient impact on the clinical efficacy of benralizumab for severe asthma ERJ, 2018, doi.1183/13993003.00936-2018)

(7) In addition mepolizumab also an anti IL5 biological shows efficacy for NP so again this would support the expectation that benralizumab would be effective (Bachert C et al ref 19) and further studies are in progress.

Author Response

Response to reviewer 3 comments

We wish to express our gratitude to the reviewer for the insightful comments. They have helped us significantly improve the paper. We have responded to the comments below.

Point 1: The patient was always eosinophilic so why wasn't aIL5 therapy used instead of BT - perhaps it would have been better to have waited until it was available

Response 1: We agree with the reviewer’s comment. The anti-IL-5 or IL-5R therapy was not available at the time of treatment with BT. He was eosinophilic, and general oral corticosteroids were needed to control asthma symptoms. He also had frequent exacerbations, which were treated with “oral corticosteroid burst.” We selected BT for this patient, as it had reduced the exacerbation rates in the AIR and AIRII trials. Currently, it would be better for this patient to start an anti-IL-5 or IL-5R therapy before BT rather than after.

We have added the following sentences in the Discussion section (L130–140 in revised manuscript): “The patient had exacerbation of asthma with hypereosinophilia six months after BT initiation, so the oral corticosteroid dose needed to be increased. During the tapering of corticosteroids, severe asthma with ECRS was diagnosed. An anti-IL-5 or IL-5Rα therapy was not available before treatment with BT. Currently, it would be better for the patient to undergo the anti-IL-5 or IL-5R therapy before BT rather than after. To treat patients with eosinophilic asthma, we should prioritize the anti-IL-5 or IL-5Rα therapy over BT.”

Point 2: Most - up to 90% of asthmatics have co-existent CRS, there was no mention of baseline assessment of CRS in this patient.

Response 2: The patient had nasal discharge with allergic rhinitis but no nasal congestion or hyposmia. There were no nasal polyps on endoscopy before treatment with omalizumab. However, the symptoms associated with ECRS were masked by the oral corticosteroids administered to treat the frequently exacerbated asthma. In accordance with the reviewer’s comment, CT of the ethmoid sinuses should be performed to diagnose ECRS at baseline.

 Point 3: Was there a reason for the acute development of hypereosinophila after BT. Could this be a complication of BT?

Response 3: In accordance with the reviewer’s comment, the hypereosinophilia after BT might be a complication of the BT treatment; however, there is no evidence of any underlying mechanism. In this case, the patient’s asthma flared-up probably because of the viral respiratory infection and induced the hypereosinophilia. Further studies are needed to explain the mechanism of hypereosinophilia after BT.

Point 4: the conclusion states that tissue eos were reduced but there was no post Rx biopsy data reported

Response 4: In accordance with the reviewer’s comment, biopsy data of the nasal polyps were added in figure 2. The images A, B, and C were obtained before benralizumab treatment, while D and E were obtained after 16 weeks and F after one year of initiation of benralizumab treatment. Accumulation of eosinophils in the nasal polyps decreased one year after the initiation of benralizumab. The eosinophil count in the nasal polyps was 215–369 per high-power field before treatment with benralizumab and 0–2 per high-power field one year after initiation of benralizumab.

 We have added the following sentences in lines 70–72: “The nasal polyps (Figure 2E) had reduced in size; moreover, eosinophilic infiltration in nasal polyps was reduced at 1 year after treatment initiation (eosinophil count in nasal polyps was 0-2 per high-power field; Figure 2F),”

 We have also added the following sentences in the legend of figure 2: “and histopathology analyses of biopsy samples from the nasal polyps. The images in A, B, and C were obtained before benralizumab treatment, while those in D and E were obtained after 16 weeks of benralizumab treatment. The image in F was obtained after 1 year of benralizumab treatment; it shows reduced accumulation of eosinophils in the nasal polyps. Benralizumab treatment resolved the ethmoid sinus shadows, reduced nasal polyp size, and decreased eosinophilic infiltration into the nasal polyps.”

Point 5: Figs 1 and 2 there is a lot of duplication in descriptive wording.

Response 5: After reflecting on this comment, we have combined figures 1 and 2 to make a new figure 1 and changed its legends.

Point 6: There is existing data to show that CRSwNP responds well to benralizumab suggesting that this is not a novel finding (Ref: Bleecker Er et al Baseline patient impact on the clinical efficacy of benralizumab for severe asthma ERJ, 2018, doi.1183/13993003.00936-2018)

Response 6: We appreciate the reviewer's concern. In the article by Bleeker Er et al., the nasal polyps were associated with greater benralizumab responsiveness for reduced exacerbation rate in patients; however, ECRS was not associated with benralizumab responsiveness for reduced exacerbation rate. Moreover, benralizumab treatment did not directly improve ECRS; in contrast, we confirmed pathologically that benralizumab depleted eosinophils in the nasal polyps after one year of its initiation (we have added the pathological findings in figures 2C and F). Thus, this was a novel finding.

Point 7: In addition mepolizumab also an anti IL5 biological shows efficacy for NP so again this would support the expectation that benralizumab would be effective (Bachert C et al ref 19) and further studies are in progress.

Response 7: In accordance with the reviewer’s comment, similar to mepolizumab, benralizumab would be effective for nasal polyps, and several clinical trials on severe chronic rhinosinusitis with eosinophilic polyposis (NCT03450083, NCT03491229) are in progress.

 We have added the following sentences in the Discussion section (L136–140): “This report demonstrated that benralizumab improved asthma control and resolved the nasal polyps and ethmoid sinus shadows on CT images of ECRS by depleting eosinophils in the peripheral blood, as well as in nasal polyp tissues. Benralizumab appears effective for nasal polyps and ECRS, and several clinical trials are underway for severe chronic rhinosinusitis with eosinophilic polyposis.”

 Thank you again for your comments on our paper. I believe that the revised manuscript is suitable for publication.
Round  2

Reviewer 1 Report

I am satisfied with the author's responses and edits.